# Improving Skin Lesion Segmentation with Self-Training

**DOI:** 10.3390/cancers16061120

**Published:** 2024-03-11

**Authors:** Aleksandra Dzieniszewska, Piotr Garbat, Ryszard Piramidowicz

**Affiliations:** Institute of Microelectronics and Optoelectronics, Warsaw University of Technology, 00-662 Warsaw, Poland; piotr.garbat@pw.edu.pl (P.G.); ryszard.piramidowicz@pw.edu.pl (R.P.)

**Keywords:** deep learning, semi-supervised learning, skin lesion segmentation, skin cancer, dermoscopy images

## Abstract

**Simple Summary:**

Finding the area of a skin lesion on dermoscopy images is important for diagnosing skin conditions. The accuracy of segmentation impacts the overall diagnosis. The quality of segmentation depends on the amount of labeled data that is hard to obtain because it requires a lot of time from experts. This study introduces a technique that enhances the segmentation process by using a combination of expert-generated and computer-generated labels. The method uses a trained model to generate labels for new data that are later used to improve the model. The findings suggest that this approach could make skin cancer detection tools more accurate and efficient, potentially making a big difference in the medical field, especially in situations where high-quality data are limited.

**Abstract:**

Skin lesion segmentation plays a key role in the diagnosis of skin cancer; it can be a component in both traditional algorithms and end-to-end approaches. The quality of segmentation directly impacts the accuracy of classification; however, attaining optimal segmentation necessitates a substantial amount of labeled data. Semi-supervised learning allows for employing unlabeled data to enhance the results of the machine learning model. In the case of medical image segmentation, acquiring detailed annotation is time-consuming and costly and requires skilled individuals so the utilization of unlabeled data allows for a significant mitigation of manual segmentation efforts. This study proposes a novel approach to semi-supervised skin lesion segmentation using self-training with a Noisy Student. This approach allows for utilizing large amounts of available unlabeled images. It consists of four steps—first, training the teacher model on labeled data only, then generating pseudo-labels with the teacher model, training the student model on both labeled and pseudo-labeled data, and lastly, training the student* model on pseudo-labels generated with the student model. In this work, we implemented DeepLabV3 architecture as both teacher and student models. As a final result, we achieved a mIoU of 88.0% on the ISIC 2018 dataset and a mIoU of 87.54% on the PH2 dataset. The evaluation of the proposed approach shows that Noisy Student training improves the segmentation performance of neural networks in a skin lesion segmentation task while using only small amounts of labeled data.

## 1. Introduction

Melanoma is the seventieth most common cancer worldwide and one of the most common cancers among young adults [1]. Although it is one of the deadliest kinds of skin cancer [2], it might be completely cured if detected early. Melanoma mortality rates are highly correlated with the state of cancer at the moment of diagnosis. Statistics show that a 5-year relative survival rate for people diagnosed in the localized stage reaches 99%, while a diagnosis in the distant stage results in a significant drop in the survival rate down to 30% [3]. Therefore, the monitoring and early diagnosis of skin lesions are crucial in preventing cancer diseases.

The majority of currently deployed solutions are focused on training complex systems toward end-use tasks such as predicting a diagnosis. These solutions have many advantages, including the computational efficiency and ease of optimization. However, excellent performance of complex models requires sufficient training data, which is often challenging in medical applications. At the same time, the performance of segmentation models has been shown to improve logarithmically with the amount of training data [4]. It is helpful to incorporate prior knowledge into the training in medical image analysis tasks with small datasets and heterogeneous case distributions. One example of this type of approach is the use of segmentation masks. This treatment reduces the complexity of understanding images by machines by extracting representative features from lesions, leading to improved diagnostic efficiency [5].

The research results presented indicate a positive correlation between enhanced segmentation quality and improved classification accuracy [6,7,8]. Our other work has demonstrated that using a segmentation mask for skin lesion classification enhances classification accuracy, and the quality of the used segmentation mask directly influences classification results [9].

A commonly used method to recognize melanoma assumes checking the ABCDE criteria. This approach considers asymmetry, border features, color, diameter, and skin lesion evolution to differentiate benign from malignant skin lesions. Approaches easily understandable by humans are commonly implemented in computer-aided diagnosis systems. They provide interpretability and explainability that end-to-end classifiers do not deliver. Doctors require computer analysis methods to not only give correct diagnoses but also explain terms on what such decision was made. The segmentation mask can provide information about the boundary and symmetry of a skin lesion. Moreover, automatic segmentation is an important preprocessing step in many medical use cases as it shows the area of interest for a further analysis. Thus, accurate skin lesion segmentation is crucial in an automated diagnosis. However, variations in shape and size, irregular lesion boundaries, and low contrast differences between the lesion and the skin make developing automated segmentation methods nontrivial.

Deep learning-based methods enable achieving state-of-the-art results in multiple medical image segmentation tasks. However, they require considerable amounts of annotated training samples, collecting which is time-consuming and costly as skilled individuals are required to label images. We propose using a semi-supervised learning technique to employ images without binary masks to improve neural network performance in a skin lesion segmentation task.

We explore the possibility of implementing self-training with Noisy Student [10] in medical image segmentation. Noisy Student training is a semi-supervised learning technique utilizing labeled and unlabeled data. It was first applied to semantic segmentation by Y. Zhu et al. [11]. It consists of three main steps. First, the teacher model is trained on a small set of labeled data, i.e., real labels. Second, labels are predicted for unlabeled data and the student model is trained using generated pseudo-labels and real labels. Third, new labels are predicted with the student model and the new student model (further referred to as student*) is trained. The self-training approach is possible in the case of skin lesion segmentation because there is an available quite large dataset of dermatoscopic skin lesion images without segmentation masks, compared to other medical image datasets. We decided to use the same architecture for the teacher and student model because there are much less public data than in a typical ImageNet classification task where millions of images are available.

Firstly, the best architecture for the teacher model was selected. Architectures like U-Net [12], U-Net++ [13], and DeepLabV3 [14] were tested. The best-performing model (DeepLabV3) was used as a teacher. The second step was to train the student model on labeled data. Then, the best student was set as a new teacher. It should be noted that the optimal ratio of labeled and unlabeled data in the training dataset was found and the further increase in the number of generated labels led to a performance decrease. Research has also shown that generated labels with better quality result in better performance of a student model. Student* slightly improves segmentation performance but only for a smaller real-to-generated labels ratio.

The evaluation of this approach shows that Noisy Student training improves the segmentation performance of neural networks in a skin lesion segmentation task while using only small amounts of labeled data. We reached state-of-the-art performance on ISIC 2018 and PH2 datasets. The code used for this research is available at https://github.com/Oichii/Improving-skin-lesion-segmentation-with-self-training (accessed on 6 March 2024).

### 1.1. Related Work

#### 1.1.1. Image Segmentation

Before deep learning grew in popularity, skin lesion segmentation methods were based on traditional image processing techniques such as adaptive thresholding based on a grayscale image histogram [15], iterative active contour adjustments [16], and region growth based on color space quantization [17].

With the rapid development of deep learning, traditional methods were replaced by convolutional neural networks. The first proposed architecture was a fully convolutional network (FCN) [18] followed by U-Net [12]. The success of encoder–decoder-type architectures led to many modifications of U-Net, like U-Net++ [13] or ResU-Net [19]. Also, different backbones were applied to improve segmentation results.

U-Net consists of a contracting path (encoder) and a symmetric expanding path (decoder). The novelty of the architecture is the concatenation of encoder intermediate feature maps with the corresponding feature maps of the decoder, enabling the network to learn context and correct localization simultaneously. The encoder follows the typical convolutional network architecture where each layer halves the input size and doubles the number of features. Each decoder layer doubles the image size and halves the number of feature channels. It is also concatenated with the appropriate feature map from the encoder. At the final layer, each feature vector is mapped to the selected number of classes [12]. DeepLab is also an encoder–decoder architecture. It utilizes dilated convolution and Atrous Spatial Pyramid Pooling (ASPP). The encoder is a convolutional network that replaces standard convolution with dilated convolution to overcome localization invariance caused by pooling operations [14]. The dilated convolution also allows the use of pre-trained weights in the encoder. DeepLab also addresses the issue of segmenting objects of varying scales through the ASPP module, which uses convolution with multiple filter sizes and dilation rates to capture multi-scale features. This approach is inspired by pyramid pooling, which showed that resampled convolutional features extracted at a single scale could correctly classify regions of any scale [20].

A specially crafted loss function combined with general-purpose architectures was applied in skin lesion segmentation. For example, a loss function based on Jaccard distance is proposed to overcome the re-weighting need [21].

Some architectures were explicitly proposed for skin lesion segmentation. A Dermoscopic Skin Network (DSNet) uses depth-wise separable convolution to eliminate the need to learn redundant features by reducing the number of parameters [22].

Wang et al. [23] proposed a boundary-aware transformer that can effectively model global long-range dependencies and capture local features by fully utilizing boundary prior knowledge provided by a boundary-wise attention gate (BAG). Because it provides detailed spatial information, BAG’s auxiliary supervision can assist transformers in learning position embedding.

Tang et al. [24] proposed the Dual-Aggregation Transformer Network (DuAT), which combines Global-to-Local Spatial Aggregation (GLSA), which, in turn, aggregates both global and local spatial features and is useful for locating objects with various scales, and Selective Boundary Aggregation (SBA), which accumulates low-level boundary characteristics and high-level semantic information for a better object localization and preservation of borders [24].

Bagheri et al. [25] proposed an ensemble of neural networks that uses a graph-based method to combine segmentation results of Mask R-CNN and Retina-Deeplab.

Input image preprocessing is also important in skin lesion image segmentation. Some recent approaches also showed that preprocessing of an input image, such as transformation to polar coordinates using the centroid or center of the object found using another method, increases skin lesion segmentation performance [26]. The proposed preprocessing method includes a hair removal technique using a black top hat filter to create a hair mask combined with an image inpainting technique to restore a clean skin image [27].

#### 1.1.2. Semi-Supervised Learning

Semi-supervised learning aims to train a model using both labeled and unlabeled data that is better than a supervised model trained on labeled data only [28]. The labeled portion of the data is usually smaller than the unlabeled portion, thus representing the most common real-life scenario. This is especially the case with medical imaging, where data collection and labeling need to be performed by a qualified doctor. The preparation of detailed masks for image segmentation is also time-consuming, even more so than for classification tasks. Consequently, segmentation benefits more from methods that allow for using only a small amount of labeled images. Semi-supervised learning can be used with both handcrafted features and deep learning-based classifiers.

You et al. [29] proposed an approach that uses self-training combined with an SVM classifier based on radial projection to segment retinal blood vessels. Portela, Cavalcanti, and Ren [30] used clustering to label voxel clusters combined with Gaussian mixture models to label the remaining pixels of a brain MR scan.

Bai et al. [31] developed an iterative semi-supervised framework for cardiac MR image segmentation where in each iteration, pseudo-labels for unlabeled images are generated by the network and refined by a conditional random field [32]. The model is then updated using generated pseudo-labels.

Adversarial learning can also incorporate unlabeled data in semi-supervised image segmentation. Zhang et al. [33] implemented two networks, one that segments images and a second that distinguishes between segmentation results of labeled and unlabeled images. In the adversarial training process, the segmentation network learns to produce similar results on both types of data.

Li et al. [34] proposed self-loop uncertainty, which involves optimizing a neural network with a self-supervised task to generate pseudo-labels, which are then used as ground truth for unlabeled images to enhance the training set and improve segmentation accuracy. This approach is a fast alternative to ensembling multiple models to estimate uncertainty as it reduces inference time.

This work is based on the Noisy Student training method introduced by Xie et al. in [10] as an extension to pseudo-labeling [35] and self-training [36]. To improve model performance, it uses unlabeled images with pseudo-labels generated by a model trained on limited labeled data. In other words, it uses the model’s own confident predictions to create more training data by producing labels for unlabeled data [28]. Image augmentations and model size also play an essential role in this approach. The student model is no smaller than the teacher to better capture the complexity of a larger dataset, and random image augmentations lead to a better generalization of the student model. It was successfully used in the segmentation task in [37] where it improved the score on PASCAL VOC 2012 and Cityscapes datasets. We found no previous use of Noisy Student training in skin lesion segmentation.

Our approach is different from other comparable solutions in the following aspects: We used deep learning instead of clustering and SVM, as proposed by You et al. [29] and Portela et al. [30]. Differently from Bai et al. [31], our pseudo-labels are generated once at the beginning of iteration and do not change during iteration. Compared to Zhang et al. [33], self-training models do not influence each other directly as in adversarial training. In our case, only pseudo-labels generated by a model are used in the next steps. We have a batch that contains labeled and unlabeled data, and we use a pseudo-label for unlabeled data. The solution proposed by Li et al. [34] also uses a batch that contains labeled and unlabeled data, but their solution uses a self-supervised subtask of image permutations for unlabeled data.

## 2. Materials and Methods

This section introduces a semi-supervised self-training framework with Noisy Student for skin lesion image segmentation. Teacher–student training, as employed in the study, is described in Figure 1. Our goal is to combine a limited set of labeled data and a large amount of unlabeled data to increase the accuracy and robustness of lesion segmentation. Such an approach allows for reducing human effort on labeling. Questions we want to answer in the study are (1) will self-training with Noisy Student enhance the segmentation of skin lesion images?; (2) what is the largest amount of unlabeled data that we can use to enhance the performance?; (3) what is the best combination of augmentations to use for input noise?

The input to the algorithm is both labeled and unlabeled images. Training a teacher model using solely labeled data is the first stage. Then, the trained teacher model predicts segmentation masks (pseudo-labels) for unlabeled images. Images with corresponding generated masks and images with real labels define a new training dataset. The teacher model generates an output image with probabilities for each pixel to belong to either the background or a skin lesion. The generated mask is then thresholded with a threshold of 0.5 to create a binary mask.

Images with generated masks that were empty or only had a small number of pixels are excluded from the dataset since they can have a negative impact on the training. In other words, the predicted confidence of skin lesion pixels was low, so the image was removed from the training dataset to increase self-training efficiency. The student model is trained to minimize the loss on both labeled and unlabeled data while validation uses only images with real labels. Finally, we run the second iteration of the training in which we select the student model with the highest IoU on the validation dataset for a new teacher. It is then employed to generate new pseudo-labels for unlabeled data. New masks are used to train a new student model (student*). The same model architecture is used for both student and teacher models so it will have enough capacity to learn from a larger dataset while preserving generalization capabilities. For student model training, we use dropout with p=0.5 as model noise and image augmentations that include random flips, rotation, and hue shift as input noise.

### 2.1. Model Architectures

In the study, we used model architectures with a notable position in the literature as we want to focus our research on designing a scalable training approach rather than on deep learning network architecture. We want to separate the influence of Noisy Student training and application-specific model adjustments. We tested four model architectures: U-Net, U-Net++, DeepLabV3, and DeepLabV3+.

U-Net was first proposed for medical image segmentation. It consists of an encoder and decoder in a U-shaped architecture. It also implements skip connections between corresponding encoder and decoder blocks, which enhance segmentation performance.

U-Net++ is an extension of the original U-Net architecture that was proposed to address some of the limitations of U-Net, particularly its limited capacity to capture complex patterns and its tendency to produce coarse segmentation results. U-Net++ takes advantage of the semantic similarity between the encoder’s and decoder’s feature maps by introducing dense skip connections. These skip connections are designed to connect each encoder layer to every layer of the corresponding decoder block. The connections also include a dense convolution block, which helps increase the network’s capacity and capture more complex patterns.

DeepLabV3 is also a commonly used solution in medical applications. It performs atrous convolution with multiple rates to capture image features at multiple scales. Model architecture is presented in Figure 2. DeepLabV3+ enhances the segmentation of object boundaries compared to DeepLabV3 by incorporating an improved decoder module.

### 2.2. Data

#### 2.2.1. Labeled Dataset

To train the baseline (teacher) model, freely accessible dermatoscopy image datasets released by the International Skin Imaging Collaboration (ISIC) were used. The combined datasets released from 2016 to 2018 contain 3074 image and segmentation mask pairs. ISIC 2016 and ISIC 2017 [38] validation and training subsets and ISIC 2018 [39,40] training subsets combined were used for model training and validation in the 4:1 ratio. The test subsets of ISIC 2016 and ISIC 2017 as well as the validation subset of ISIC 2018 were used to test the model. Acquired subsets were checked to ensure no overlap between the data. In total, there are 1572 training, 523 validation, and 979 testing mask and image pairs.

For the final evaluation, we used the PH2 dataset that contains 200 dermoscopic images with segmentation masks.

#### 2.2.2. Unlabeled Dataset

Unlabeled images were also obtained from freely accessible dermatoscopy image datasets, i.e., ISIC 2020 [41,42] and ISIC 2019 [43] datasets. Those datasets combined provide almost 60k skin lesion images without segmentation masks. The number of samples in each of the datasets is presented in Figure 3; for the training dataset, we used a maximal number of pseudo-labels in the study, i.e., for ratio m=8. Images from the labeled datasets were filtered out from the unlabeled data. Then, the model trained on labeled data was run on the images to predict segmentation masks. Masks with a pixel size of less than 100 were screened. This was performed because the masks containing only a few pixels are less accurate or contain errors. The examples of rejected images are shown in Figure 4.

### 2.3. Implementation Details

We implemented the method described above using the PyTorch framework [44]. Encoder models were initialized with weights pre-trained on ImageNet, and decoder weights were initialized randomly. Pre-trained weights are used due to their beneficial influence on skin lesion segmentation performance, as it was shown in [45]. Images and masks were resized to the resolution of 256 × 256 pixels, and values were scaled to the range of [0,1]. We used a batch size of bs=10 by default and reduced it when we could not fit the model into the memory.

For training, a stochastic gradient descent optimizer (SGD) [46] was used with an initial learning rate of lr=0.002 and cosine annealing learning rate scheduler [47]; momentum was set to β=0.54 and weight decay was set to wd=0.01.

We trained each DeepLabV3 with a ResNet18 backbone for 60 epochs and DeepLabV3 with ResNet34 for 90 epochs or until IoU on the validation dataset no longer decreased.

As a loss function, dice loss presented in Equation (Equation 1) was used, where λ=1 is the smoothing parameter. It provides better results in terms of both the IoU and dice coefficient compared to the weighted cross-entropy function.
(1)$$L_{Dice}=1-\frac{2|X⋂Y|+\lambda }{|X|+|Y|+\lambda }$$

### 2.4. Evaluation Metrics

Model performance was evaluated with the dice coefficient (*Dice*), presented in Equation (Equation 2), and Intersection over Union (*IoU*), also known as the Jaccard index, presented in Equation (Equation 3), where *X* is the predicted mask and *Y* is the ground truth mask. The Jaccard index is used to quantify the overlap area between the true and predicted lesion masks, and the dice coefficient is used to assess the similarity between real and predicted masks.
(2)$$Dice=\frac{2|X⋂Y|}{\left|X|+|Y\right|}$$
(3)$$IoU=\frac{\left|X⋂Y\right|}{\left|X⋃Y\right|}$$

In addition, precision (*Prec*) and recall (*Rec*) were calculated in a pixel-wise manner as in Equations (Equation 4) and (Equation 5), where TP is the number of true positives, FP is the number of false positives, and FN is the number of false negatives. Precision is used to measure the number of pixels that belong to the lesion region that are correctly classified. Recall calculates the number of pixels outside of the lesion area that are incorrectly classified.
(4)$$Prec=\frac{TP}{TP+FP}$$
(5)$$Rec=\frac{TP}{TP+FN}$$

All evaluation metrics take values in the range [0,1] and the higher values correspond to better results. Following the approach from work [27], the percentage of images with IoU over 0.8 was calculated and a visual inspection of the worst and best examples was performed. It is due to the expert dermatologists’ agreement that only skin lesion segmentation with IoU over 78.6% is helpful and useful for medical purposes [40]. Also, segmentation with the Jaccard index equal to 0.8 or above is, in general, visually correct [38].

## 3. Results

In this section, we describe the details of our experiments. Then, we report the method’s performance on ISIC 2018 and PH2 datasets as they are the most benchmarked and freely accessible skin lesion segmentation datasets. Finally, we compare our method with the state-of-the-art skin lesion segmentation models described in the literature.

### 3.1. Architecture Selection for Teacher Model

We compared U-Net, U-Net++, DeepLabV3, and DeepLabV3+ architectures to determine the optimal architecture for skin lesion segmentation, and the best one was selected for further experiments. All the models were tested with a ResNet34 backbone. Results are shown in Table 1.

U-Net++ provides the highest dice coefficient and recall, and DeepLabV3 produces the most promising IoU and precision. We decided to optimize our model for an optimal Jaccard index so for further experiments, a model with the best IoU on a test set containing combined images from ISIC 2017 + 2018 datasets was selected. We assumed the same network architecture for teacher and student models. Thus, in the following, unless stated otherwise, the baseline for all of our experiments is DeepLabV3 with a ResNet34 backbone, which achieved a mIoU of 82.05%.

### 3.2. Real- to Pseudo-Label Ratio

We proposed an experiment to find the optimal composition of the dataset for training the student model. It will determine if adding more unlabeled data has a positive influence on segmentation results. The training set consisted of n=1572 images with real labels and n×m pseudo-labels where m∈{1,2,4,8}. The validation set was the same in each experiment and included 523 images with real labels.

Results of training models described in Section 2 for the ISIC 2018 dataset are shown in Table 2 for the ResNet18 backbone and Table 3 for the ResNet34 backbone.Results for the PH2 dataset are shown in Table 4 for the ResNet18 backbone and Table 5 for the ResNet34 backbone. For both backbones, the addition of unlabeled data enhances segmentation results in all metrics compared with the teacher model. However, ratios higher than m=4 might decrease performance compared with smaller ratios, as shown in Figure 5.

### 3.3. Input Noise

Image augmentations are a key part of this study as they address two problems—overfitting and input noise for student training. The overfitting appears during the teacher training process while using limited labeled data. In this case, augmentations make the model more robust for unseen data. A controlled input noise level enhances the results of semi-supervised training because it enforces consistency of the decision on labeled and unlabeled data [10]. It is due to the fact that the student must replicate the high-quality pseudo-label that the teacher created on the original version of the image on the augmented noisy version [48].

We implemented simple, commonly used augmentations, which include random brightness, contrast and saturation adjustments, blur, the rotation of a random angle, and horizontal and vertical flips as a baseline. Moreover, we examined the influence of modern augmentations, i.e., coarse dropout, optical distortion, elastic transform, and grid distortion, on segmentation performance. We combined each modern augmentation with a set of simple augmentations. Figure 6 shows the described sets of used augmentations.

Table 6 compares teacher and student models trained with different augmentation sets and values of the real- to pseudo-label ratio in training dataset *m* for which the student model was best-performing. The use of grid distortion results in a better teacher model, but the performance of student models decreases rapidly with higher ratios of pseudo-labeled data in the training dataset. Coarse dropout results in the weakest teacher model but gives the highest increase in performance from teacher to student. However, improvement from teacher to student is still insufficient compared to other models (see Figure 7). Optical distortion provides good performance of the teacher model and a slighter decrease in performance with higher pseudo-labeled data ratios in the training dataset, so it was selected for further experiments. Results are shown in Table 7 and Table 8.

### 3.4. Second Iteration of Student Training

The best-performing models from previous experiments were used for the second iteration of training. Using these models, new pseudo-labels were generated for unlabeled data in place of pseudo-labels used in the previous iteration. The dataset for student* training also contains real- and pseudo-labels with different ratios *m* as in the previous experiment.

Figure 8 shows progress between teacher, student, and student* models. Progress from teacher to student is significant, but the second iteration of Noisy Student training does not bring considerable improvement. This dependence is more visible for a model with a ResNet34 backbone, which started from a better teacher. In this case, growth from teacher to student is only 0.78% while the model with a ResNet18 backbone that started with a worse teacher enhanced its performance by 2.93%.

The introduction of advanced augmentation, i.e., optical transform, leads to better performance of the second iteration of Noisy Student training. This effect is shown in Figure 9. The use of optical distortion leads to significant accuracy improvement between student and student* models compared to the model with simple augmentations.

Progress of each iteration of self-training is shown in Figure 10 on the ISIC 2018 dataset and Figure 11 on the PH2 dataset. An increase in performance is still visible on those datasets for each configuration. For the student model with a ResNet34 backbone and simple augmentations, better performance on the ISIC 2018 dataset was achieved for models with an m=2 ratio as shown in Figure 5; nonetheless, the model with m=4 was used as a teacher based on performance on the ISIC 2017 + 2018 dataset.

Figure 12 shows a comparison of student and student* models for each ratio of real- to pseudo-labels. The student* model improves only in a small ratio of real- to pseudo-labels and for a bigger ratio performance decreases below the teacher model. The higher the ratio, the higher the decrease in performance. Higher ratios of 1:4 and 1:8 decline performance even below the new teacher model for the same ratio. Results on the ISIC 2018 validation subset are presented in Table 9 and Table 10 and for the PH2 dataset in Table 11 and Table 12. Results with optical transform are shown in Table 13 (ISIC 2018) and Table 14 (PH2). We also ran a test with test time augmentations to verify that our model has satisfactory generalization capabilities. We applied random augmentations including flips, shifts, brightness and contrast adjustments, hue shifts, histogram equalization, and rotation to PH2 data and achieved an IoU of 0.8704, a precision of 0.8801, and a recall of 0.9611 (mean of 10 runs with random augmentations), which is similar to the described test results on the original dataset shown in Table 11.

The PH2 dataset was used for the statistical analysis. For the teacher model, we achieved IoU of 0.87 with a 95% confidence interval of [0.84,0.89], precision of 0.90 with a 95% confidence interval of [0.88,0.91], recall of 0.94 with a 95% confidence interval of [0.92,0.95]. For the student* model, we achieved IoU of 0.88 with a 95% confidence interval of [0.85,0.89], precision of 0.89 with a 95% confidence interval of [0.87,0.90], recall of 0.97 with a 95% confidence interval of [0.95,0.98].

## 4. Discussion

From the experiments presented above, it is clearly seen that the introduction of unlabeled data leads to a performance increase in the student model above the baseline teacher model in all presented configurations of network architecture and input noise. Figure 5 and Figure 7 show the influence of the composition of the training set on segmentation performance measured by mIoU on the combined ISIC 2017 + 2018 test dataset. Increasing the pseudo-labeled data number in the dataset results in the enhancement in segmentation only to a certain extent. The optimal ratio lies between 1:2 and 1:4 depending on augmentations and the backbone used. The best-performing configuration, which is DeepLabV3 with a ResNet34 backbone with an optical distortion augmentation, achieves a mIoU value of 85.49% for the ratio of 1:2; further increasing the pseudo-labeled data number in the training dataset leads to a significant performance decline. The second promising configuration, which is DeepLabV3 with a ResNet34 backbone with simple augmentations, has the optimal segmentation performance when the dataset consists of 4/5 of pseudo-labeled and 1/5 of real-labeled data. This configuration achieves a mIoU value of 85.40%. A higher ratio of real- to pseudo-labels decreases the performance of the student model. Different input noise was tested in Section 3.3, and the most suitable turned out to be optical transform. This configuration gives the most robust model on combined ISIC 2017 and 2018 datasets.

In terms of the backbone, ResNet18 is insufficient for the task due to a smaller amount of parameters, so it cannot learn more complex boundary information. This leads to worse performance compared to ResNet34, which can capture skin lesion texture and boundaries better as shown in Figure 5.

Results show that the better the teacher model, the better performance the student model will achieve. On the other side, better teacher performance results in a smaller increase in student performance. The teacher model that has a higher accuracy allows for achieving better results in Noisy Student training. A possible reason for this can be that the ResNet34 backbone allows predicting the pseudo-labels more accurately than the ResNet18 model. The incorrect labels will make the student model learn the wrong segmentation labels, which leads to a decrease in the effectiveness of the learner model.

Results on the ISIC 2018 dataset are better than on the full test subset containing test sets from all ISIC segmentation challenges because images in the 2018 dataset are less diverse and have better annotations tight to the skin lesion boundary so they are more in line with the model’s specification. Conversely, the smaller datasets will better fit models with fewer parameters. Therefore, for the ISIC 2018 dataset, we can see the advantage of the model with the ResNet18 backbone over the model with the ResNet34 backbone. A smaller model allows for better generalization with a limited number of training examples, which prevents overfitting during training. This smaller number of model parameters helps the student model learn better from fewer images.

Analyzing the distribution of IoU scores is just as significant as investigating the mean IoU on the test set as it shows how many images are below the threshold of clinical relevance. Figure 13 shows the distribution of IoU scores for images in the test set. For the student* model, 83.0% of images achieved an IoU value above 0.8 on the ISIC dataset and 86.5% on the PH2 dataset.

### 4.1. Qualitative Analysis

Figure 14 shows images with the smallest IoU for teacher, student, and student* models compared with ground truth masks. The main reasons for segmentation failures are small skin lesions with low contrast between healthy skin and lesion tissues. Another typical failure case was subjectively incorrect annotation not tight to the skin lesion boundary. In the case of incorrect ground truth, the model predicted masks that resembled actual lesion shapes.

Figure 15 shows images with the highest IoU achieved by the student* model. This case shows that the student* model learns to segment detailed boundaries better even though the teacher model only roughly outlined the shape of the lesion.

### 4.2. Comparison with State of the Art

Table 15 shows the comparison with other skin lesion segmentation methods published in recent years representing state-of-the-art results. We compare models on the ISIC 2018 dataset, as this is the dataset used in most publications to report results. Our presented model is DeepLabV3 with a ResNet18 backbone student* model as it achieved the best performance on this dataset.

Presented top-performing methods are end-to-end networks [23,24,49], or multistage segmentation methods [26], but they do not utilize any additional unlabeled data to increase segmentation performance. We have shown that by using simple, general-purpose architecture and self-training, it is possible to outperform complex, specifically tailored methods.

### 4.3. The Robustness of the Model

The problem of the robustness of a model for skin cancer segmentation has been discussed in many publications [50]. It depends on various factors, including its architecture, training data, generalization ability, and performance across different datasets and scenarios.

Augmenting the training data with techniques like transposition, vertical and horizontal flip, brightness, contrast, hue adjustments, CLAHE, shifts, rotation, and coarse dropout improves the model’s ability to generalize to unseen data and enhance its robustness. The results of experiments with test time augmentations on the PH2 dataset, i.e., IoU=0.87, Prec=0.88, Recall=0.96, confirm that the proposed solution maintains high accuracy even with minor variations in the input data.

The metrics proposed in the validation experiment provide insights into the model’s performance and help identify potential weaknesses. For the student* model, we notice the increasing precision and recall parameters. Calculated confidence intervals for evaluation metrics to quantify the uncertainty in the model’s performance estimates, presented in Section 3.4, confirm the high robustness of the proposed solution.

The models were evaluated on unseen data from different sources to assess their generalization ability. The PH2 dataset was used in the final evaluation. Our models generalize well to diverse datasets and imaging conditions beyond the training distribution. The obtained evaluation metrics for the PH2 dataset remain at the same level as those calculated for the ISIC dataset. The low variability of the result suggests good generalization performance.

## 5. Conclusions

In this work, we introduced and discussed the self-training framework for skin lesion segmentation. The approach is based on iterative model training and generating new labels for available unlabeled data. The self-training strategy can use vast amounts of unlabeled data to increase the accuracy of the segmentation model. Experiments have shown that the addition of unlabeled data leads to performance improvement in all tested configurations. We performed a quantitative and qualitative analysis of model performance, which shows that the proposed model yields state-of-the-art results in skin lesion segmentation tasks on two skin lesion segmentation benchmark datasets, ISIC 2018 and PH2. Finally, we achieved a mIoU value of 88.0% on ISIC 2018 and 87.5% on PH2 datasets. Such results were acquired for the second iteration of Noisy Student training in which pseudo-labels were generated from a model trained on real- and pseudo-labels. The study’s main contribution is to prove that a simple network with self-training can outperform a complex network in a skin lesion segmentation task. In the future research, we plan to investigate the influence of self-training on application-specific model architecture. Additionally, we found the optimal composition of the training dataset and the most suitable augmentation set for training data to achieve optimal performance of the student model. When the ratio of real- to pseudo-labels is too high, performance of the models starts to decrease in some cases even below the teacher model level. We have shown that a better teacher model, thereby pseudo-labeled data of better quality, results in better performance of the student model.

## Figures and Tables

**Figure 1 cancers-16-01120-f001:**
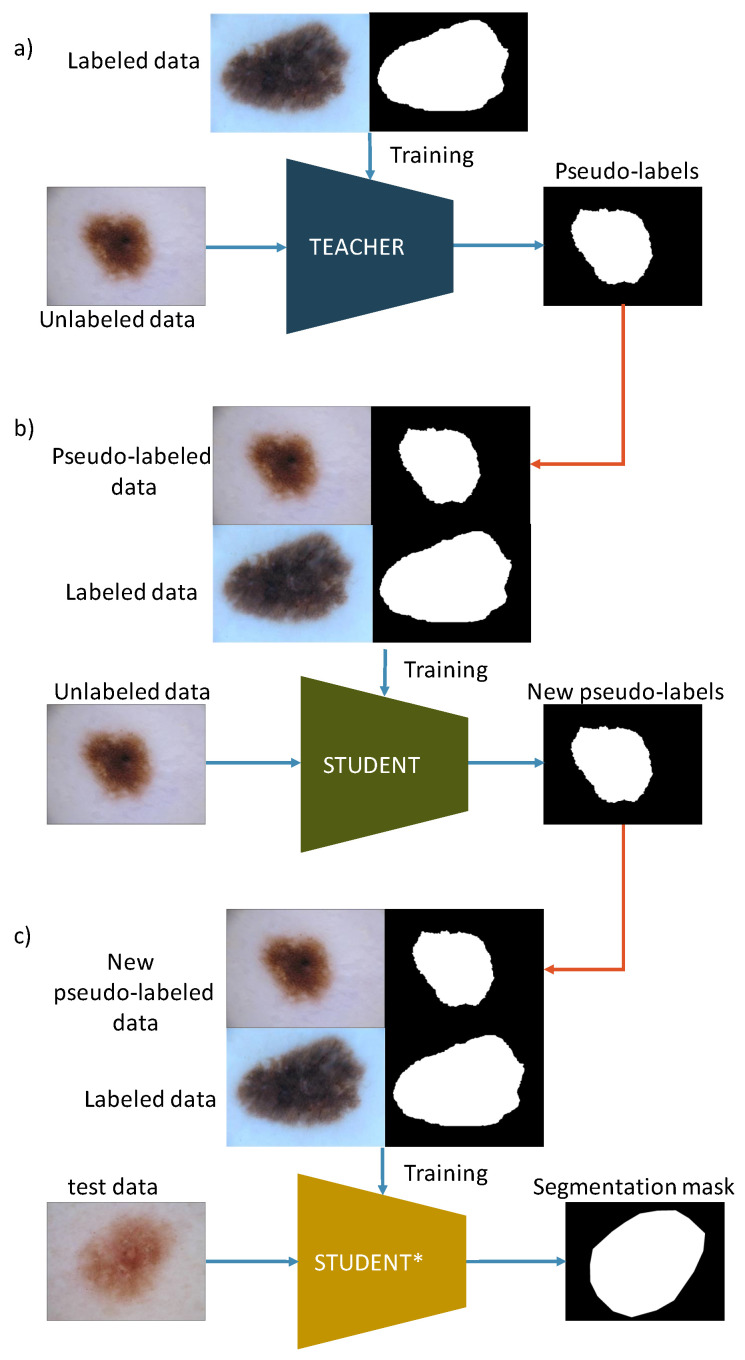
Self-training with Noisy Student for skin lesion image segmentation. (**a**) Train teacher model on labeled data and generate pseudo-labels. (**b**) Train the student model on labeled and pseudo-labeled data and generate new pseudo-labels. (**c**) Train the student* model on labeled and new pseudo-labeled data. Blue arrows in the Figure represent the model training process, and red arrows describe pseudo-labels flow.

**Figure 2 cancers-16-01120-f002:**
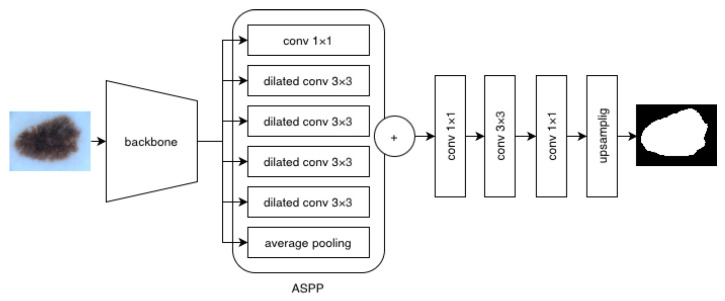
DeepLab V3 model architecture [14].

**Figure 3 cancers-16-01120-f003:**
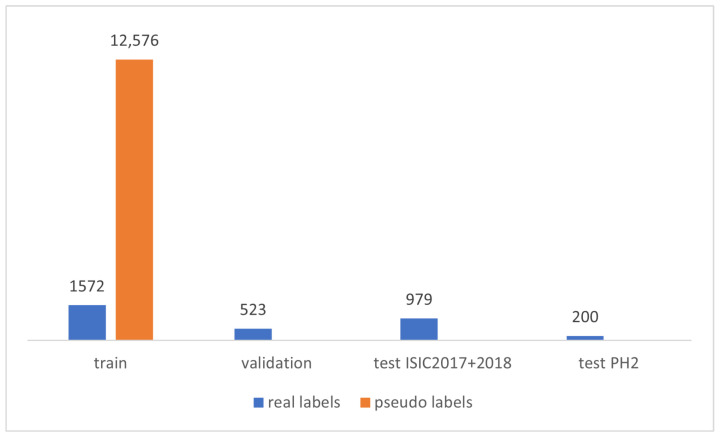
Number of real and pseudo-labeled samples in train, validation, and test datasets.

**Figure 4 cancers-16-01120-f004:**
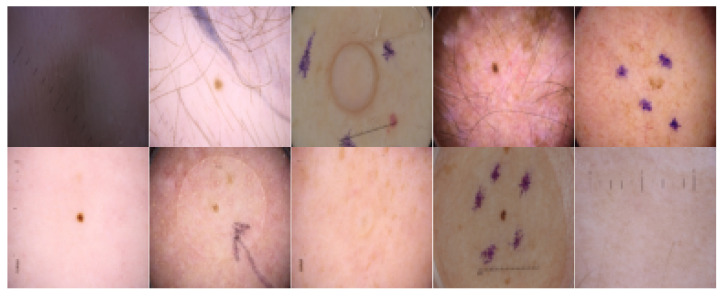
Examples of images for which masks were not created or were too small and thus not included in the dataset.

**Figure 5 cancers-16-01120-f005:**
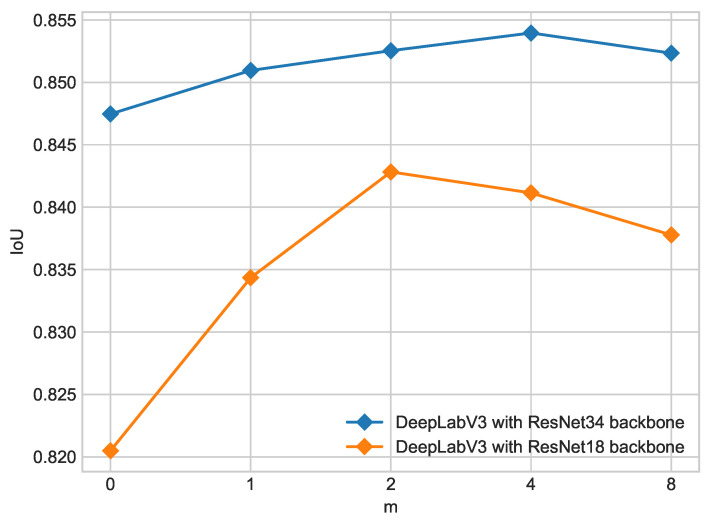
Influence of training dataset composition on IoU for different backbones on ISIC 2017 + 2018 dataset. Ratio of labeled to unlabeled data is 1/*m* where m=0 denotes teacher model.

**Figure 6 cancers-16-01120-f006:**
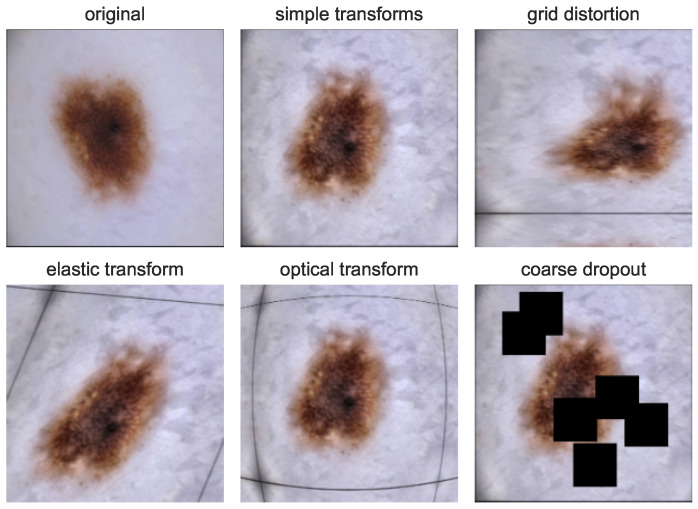
Original skin lesion image and its augmented versions. Each shows the augmentation set used in the study as input noise for training student model.

**Figure 7 cancers-16-01120-f007:**
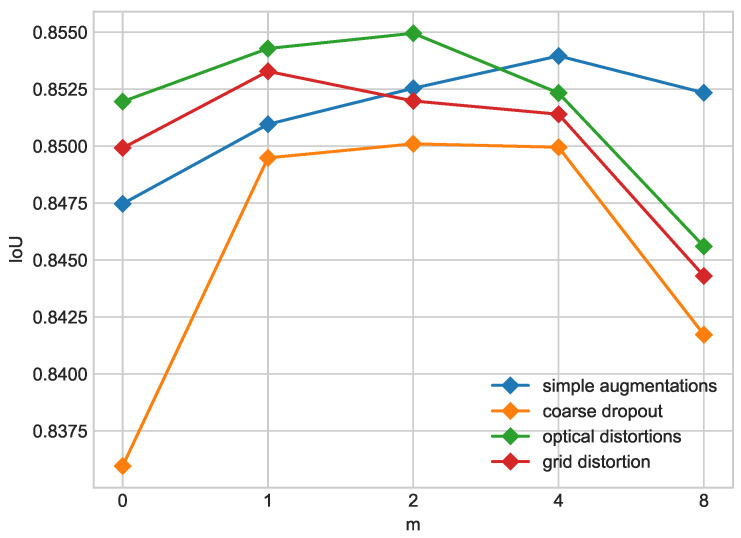
Influence of real- to pseudo-label ratio on IoU for different augmentation sets on ISCI 2017 + 2018 dataset. Ratio of labeled to unlabeled data is 1/*m* where m=0 denotes teacher model.

**Figure 8 cancers-16-01120-f008:**
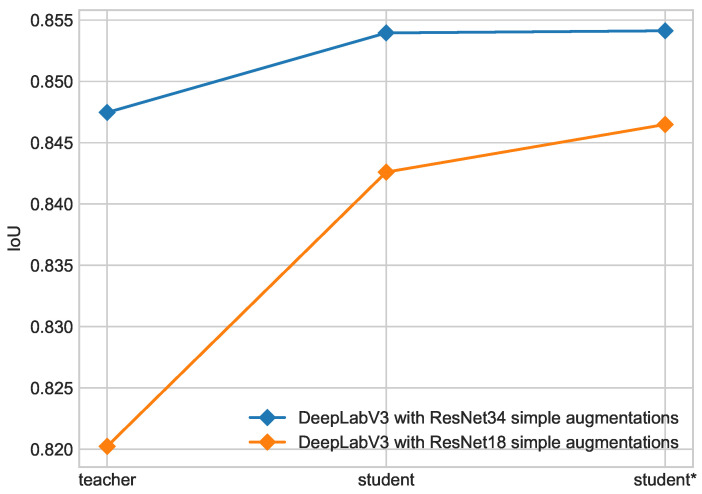
Progress in each iteration of self-training for DeepLabV3 with ResNet18 backbone and DeepLabV3 with ResNet34 backbone on ISIC 2017 + 2018 dataset.

**Figure 9 cancers-16-01120-f009:**
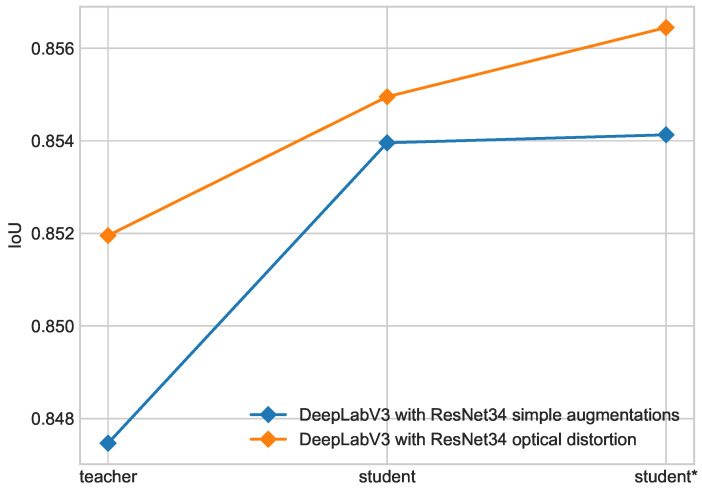
Progress in each iteration of self-training for DeepLabV3 with ResNet34 backbone with simple augmentations and optical transform on ISIC 2017 + 2018 dataset.

**Figure 10 cancers-16-01120-f010:**
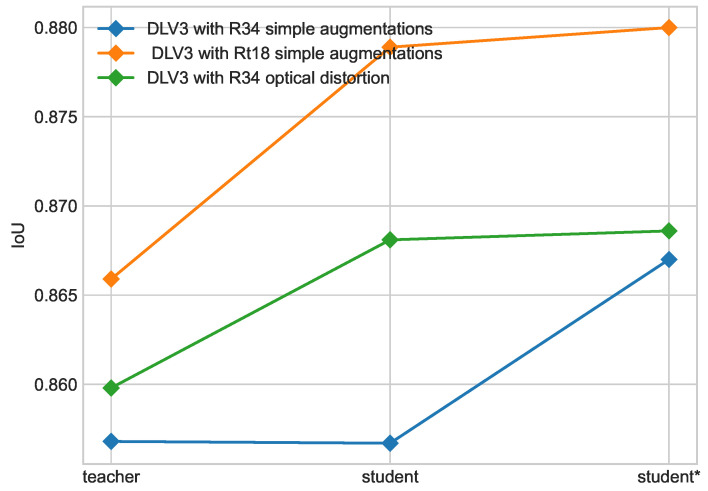
Progress in each iteration of self-training on ISIC 2018 dataset. Results for DeepLabV3 (DLV3) with ResNet34 (R34) and ResNet18 (R18) backbones.

**Figure 11 cancers-16-01120-f011:**
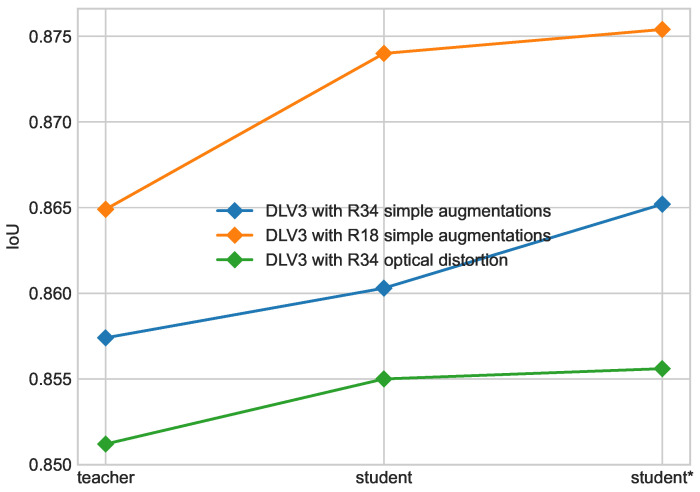
Progress in each iteration of self-training on PH2 dataset. Results for DeepLabV3 (DLV3) with ResNet34 (R34) and ResNet18 (R18) backbones.

**Figure 12 cancers-16-01120-f012:**
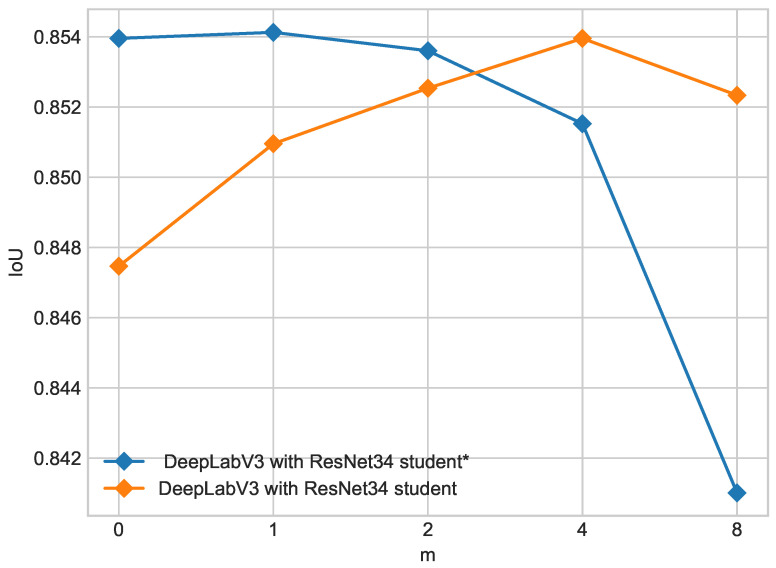
Influence of real- to pseudo-label ratio on student and student* models on ISIC 2017 + 2018 dataset. The ratio of labeled to unlabeled data is 1/*m*, where m=0 denotes the teacher model. For student*, IoU improves only for 1:1 real- to pseudo-labels, and for higher ratios, performance starts to decrease, while for the student model, optimal performance is for a 1:4 ratio.

**Figure 13 cancers-16-01120-f013:**
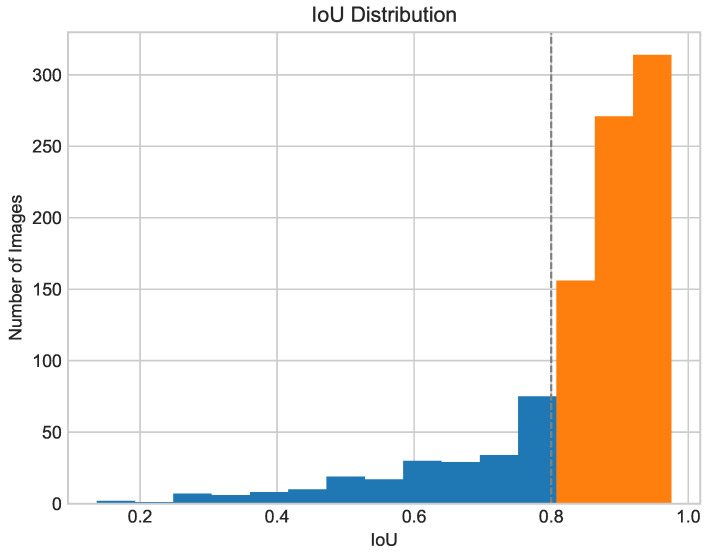
Distribution of IoU in ISIC 2017 + 2018 test set. The dashed line marks IoU=0.8 as threshold of correct segmentation.

**Figure 14 cancers-16-01120-f014:**
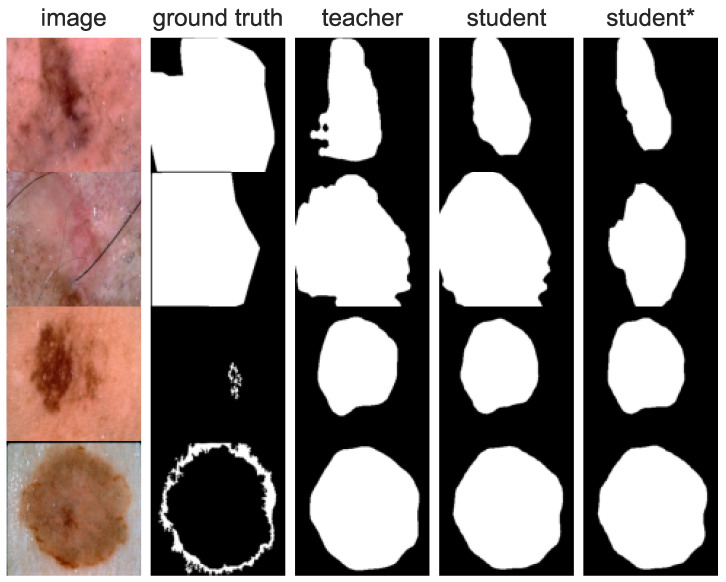
Images with the worst acquired IoU scores. Subjective assessment of some ground truth masks shows that images are annotated incorrectly. Nevertheless, the model predicted masks that resembled actual lesions.

**Figure 15 cancers-16-01120-f015:**
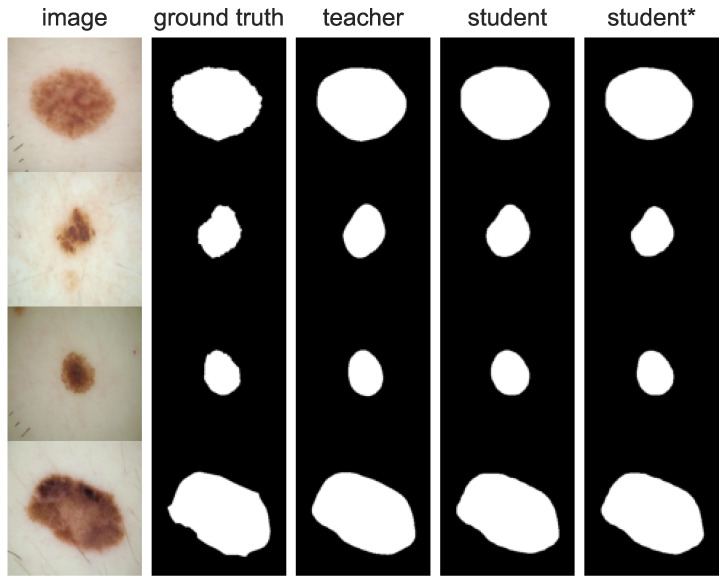
Images with the best acquired IoU scores. We demonstrate that each iteration of self-training leads to more detailed skin lesion boundary segmentation.

**Table 1 cancers-16-01120-t001:** Comparison of model architectures for skin lesion segmentation.

	IoU	Dice	Precision	Recall
U-Net	0.6941	0.7396	0.8758	0.6460
U-Net++	0.8137	0.8978	0.8902	0.8202
DeepLabV3	0.8205	0.8822	0.9170	0.8080
DeepLabV3+	0.8166	0.8843	0.9140	0.7915

**Table 2 cancers-16-01120-t002:** Results of student with simple augmentations model (ResNet18 backbone) depending on train set composition on ISIC 2018 dataset.

	mIoU	Dice	Precision	Recall
teacher	0.8659	0.9194	0.8781	0.9308
*m* = 1	0.8789	0.9355	0.9030	0.9303
*m* = 2	0.8713	0.9215	0.8690	0.9482
*m* = 4	0.8657	0.9245	0.8760	0.9382
*m* = 8	0.8714	0.9292	0.8808	0.9396

**Table 3 cancers-16-01120-t003:** Results of student with simple augmentations model (ResNet34 backbone) depending on train set composition on ISIC 2018 dataset.

	mIoU	Dice	Precision	Recall
teacher	0.8568	0.9133	0.8408	0.9596
*m* = 1	0.8647	0.9173	0.8498	0.9561
*m* = 2	0.8601	0.9159	0.8622	0.9389
*m* = 4	0.8567	0.9180	0.8368	0.9579
*m* = 8	0.8593	0.9124	0.8448	0.9571

**Table 4 cancers-16-01120-t004:** Results of student with simple augmentations model (ResNe18 backbone) depending on train set composition on PH2 dataset.

	mIoU	Dice	Precision	Recall
teacher	0.8649	0.9273	0.8962	0.9385
*m* = 1	0.8740	0.9382	0.8969	0.9503
*m* = 2	0.8723	0.9333	0.8816	0.9652
*m* = 4	0.8578	0.9224	0.8590	0.9671
*m* = 8	0.8734	0.9346	0.8877	0.9590

**Table 5 cancers-16-01120-t005:** Results of student with simple augmentations model (ResNet34 backbone) depending on train set composition on PH2 dataset.

	mIoU	Dice	Precision	Recall
teacher	0.8574	0.9192	0.8502	0.9760
*m* = 1	0.8554	0.9181	0.8459	0.9782
*m* = 2	0.8575	0.9200	0.8475	0.9791
*m* = 4	0.8603	0.9222	0.8484	0.9811
*m* = 8	0.8537	0.9135	0.8380	0.9825

**Table 6 cancers-16-01120-t006:** IoU on ISIC 2017 + 2018 of DeepLabV3 with ResNet34 backbone with different augmentations. *m* stands for real- to pseudo-labeled data in the student training dataset.

	Teacher	Student	*m*
simple augmentations	0.8475	0.8540	4
coarse dropout	0.8360	0.8501	2
elastic transform	0.8169	0.8299	1
grid distortion	0.8499	0.8532	1
optical distortion	0.8519	0.8550	2

**Table 7 cancers-16-01120-t007:** IoU on PH2 of DeepLabV3 with ResNet34 backbone with optical distortion augmentation. *m* stands for real- to pseudo-labeled data in the student training dataset.

	mIoU	Dice	Precision	Recall
teacher	0.8512	0.9226	0.8451	0.9820
*m* = 1	0.8586	0.9253	0.8536	0.9804
*m* = 2	0.8550	0.9222	0.8454	0.9843
*m* = 4	0.8546	0.9217	0.8473	0.9806
*m* = 8	0.8484	0.9227	0.8425	0.9826

**Table 8 cancers-16-01120-t008:** IoU on ISIC 2018 of DeepLabV3 with ResNet34 backbone with optical distortion augmentation. *m* stands for real- to pseudo-labeled data in the student training dataset.

	mIoU	Dice	Precision	Recall
teacher	0.8598	0.9209	0.8404	0.9710
*m* = 1	0.8559	0.9207	0.8470	0.9603
*m* = 2	0.8681	0.9220	0.8596	0.9590
*m* = 4	0.8657	0.9245	0.8760	0.9382
*m* = 8	0.8439	0.9104	0.8367	0.9477

**Table 9 cancers-16-01120-t009:** Results of teacher, student, and student* on ISIC 2018 dataset (ResNet18 backbone).

	mIoU	Dice	Precision	Recall
teacher	0.8659	0.9194	0.8781	0.9308
student	0.8789	0.9355	0.9030	0.9303
student*	0.8800	0.9373	0.9002	0.9342

**Table 10 cancers-16-01120-t010:** Results of teacher, student, and student* on ISIC 2018 dataset (ResNet34 backbone).

	mIoU	Dice	Precision	Recall
teacher	0.8568	0.9133	0.8408	0.9596
student	0.8567	0.9180	0.8368	0.9579
student*	0.8670	0.9213	0.8593	0.9565

**Table 11 cancers-16-01120-t011:** Results of teacher, student, and student* on PH2 dataset (ResNet18 backbone).

	mIoU	Dice	Precision	Recall
teacher	0.8649	0.9273	0.8962	0.9385
student	0.8740	0.9382	0.8969	0.9503
student*	0.8754	0.9372	0.8858	0.9651

**Table 12 cancers-16-01120-t012:** Results of teacher, student, and student* on PH2 dataset (ResNet34 backbone).

	mIoU	Dice	Precision	Recall
teacher	0.8574	0.9192	0.8502	0.9760
student	0.8603	0.9222	0.8484	0.9811
student*	0.8652	0.9259	0.8554	0.9810

**Table 13 cancers-16-01120-t013:** Results of teacher, student, and student* on ISIC 2018 dataset (ResNet34 backbone with optical distortion augmentation).

	mIoU	Dice	Precision	Recall
teacher	0.8598	0.9209	0.8404	0.9710
student	0.8681	0.9220	0.8596	0.9590
student*	0.8686	0.9189	0.8632	0.9545

**Table 14 cancers-16-01120-t014:** Results of teacher, student, and student* on PH2 dataset (ResNet34 backbone with optical distortion augmentation).

	mIoU	Dice	Precision	Recall
teacher	0.8512	0.9226	0.8451	0.9820
student	0.8550	0.9222	0.8454	0.9843
student*	0.8556	0.9182	0.8404	0.9804

**Table 15 cancers-16-01120-t015:** Segmentation performance metrics for the proposed method and other state-of-the-art methods on ISIC 2018 dataset. Note that the Polar method authors reported median IoU (result marked with *).

Method	mIoU	Dice	Ref.
DuAT	0.867	0.923	[24]
BAT	0.843	0.912	[23]
DoubleU-Net	0.821	0.896	[49]
Polar	0.874 *	0.925	[26]
Ours (ResNet18)	0.880	0.937	[this work]
Ours (ResNet34)	0.869	0.919	[this work]

## Data Availability

Links to datasets used in the study: ISIC datasets https://challenge.isic-archive.com/data/ (accessed on 15 October 2022), PH2 dataset https://www.fc.up.pt/addi/ph2%20database.html (accessed on 15 October 2022).

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
