# Peer review of "Improving Skin Lesion Segmentation with Self-Training"

_cancers, 2024, doi:10.3390/cancers16061120_

Round 1

Reviewer 1 Report

Comments and Suggestions for Authors

The author proposed a new semi-supervised skin lesion segmentation method using self-training with noisy students. The DeepLabV3 architecture was implemented as both teacher and student models. The method achieves an mIoU of 88.0% on the ISIC 2018 dataset and 87.54% on the PH2 dataset. For better publication, my suggestions are as follows:

1. It is recommended that the author conduct a detailed comparison between the performance of this research model and previous studies in the Related Work or Discussion section, which can be presented in the form of a list.

2. In the “2.1. Data” section, it should be detailed about the distribution of all data components of this study, which can be presented in the form of charts.

3. In "2.1.1. Labeled dataset", "to train the baseline (teacher)... " should be "To train the baseline (teacher)...". In "2.1.2. Unlabeled dataset", "unlabeled images also were obtained... " should be "Unlabeled images also were obtained...".

4. It is recommended to supplement the structural diagram of the model and provide detailed descriptions.

5. It is recommended to supplement the detailed process of model training.

6. It is recommended to discuss the robustness of the model to these common problems.

Author Response

Thank you for taking the time to review our manuscript. We appreciate your accurate comments and suggestions to improve our manuscript. Please find the detailed responses to each comment; see also the attachment for responses in table form.

Comment 1: It is recommended that the author conduct a detailed comparison between the performance of this research model and previous studies in the Related Work or Discussion section, which can be presented in the form of a list.

Ad.1 We compared our work with other skin lesion segmentation methods presented in the literature in Table 15 (page 20). For the comparison, results extracted from the papers were used. We also added a detailed comparison of our approach with other semi-supervised methods presented in the literature review (section 1.1.2. Semi-supervised learning).

Our approach is different from other comparable solutions in the following aspects. We used deep learning instead of clustering and SVM, as proposed by You et al. and Portela et al. Differently from Bai et al., our pseudo labels are generated once at the beginning of iteration and do not change during iteration. Compared to Zhang et al., self-training models do not influence each other directly as in adversarial training. In our case, only pseudo labels generated by a model are used in the next steps. We have a batch that contains labeled and unlabeled data, and we use pseudo-label for unlabeled data. The solution proposed by Li et al. also uses a batch that contains labeled and unlabeled data, but their solution uses a self-supervised subtask of image permutations for unlabeled data. (lines 190 - 198)

Comment 2: In the “2.1. Data” section, it should be detailed about the distribution of all data components of this study, which can be presented in the form of charts.

Ad. 2 We added Figure 3 (page 8), which shows the numbers of real and pseudo-labeled samples in train, validation, and test datasets.

Comment 3: In "2.1.1. Labeled dataset", "to train the baseline (teacher)... " should be "To train the baseline (teacher)...". In "2.1.2. Unlabeled dataset", "unlabeled images also were obtained... " should be "Unlabeled images also were obtained...".

Ad. 3 We capitalized the first letters of the mentioned sentences (lines 252 and 263).

Comment 4: It is recommended to supplement the structural diagram of the model and provide detailed descriptions.

Ad. 4 We added a diagram presenting model architecture (Figure 2, page 7).

Comment 5: It is recommended to supplement the detailed process of model training.

Ad. 5 We will make our training codes publicly available when the paper is published.

Codes will be available on GitHub (https://github.com/Oichii/Improving-skin-lesion-segmentation-with-self-training). We also described the training process in section 2.3, Implementation details. We provided information on used image preprocessing, hyperparameters, optimizers, batch sizes, training length, and loss function.

Comment 6: It is recommended to discuss the robustness of the model to these common problems.

Ad. 6

The robustness of a skin lesion segmentation model depends on various factors, including its architecture, training data, generalization ability, and performance across different datasets and scenarios. The models were evaluated on unseen data from different sources to assess their generalization ability. To ensure that the presented results are reliable, we maintained a test subset of data that was not used for training and validation; we also tested our model on an entirely new dataset (PH2) that was not included in the training data and came from a different source than the dataset used for training. We ran an additional test with test time augmentations to prove that our model has good generalization. It is to introduce additional controlled perturbations to the input images from the test set and measure the impact on the model's performance. The proposed solution maintains high accuracy even with minor variations in the input data. We also calculated the 95% confidence intervals on the PH2 dataset.

We added appropriate fragments in the manuscript:

In subsection 3.4. Second iteration of student training:

We also ran a test with test time augmentations to verify that our model has satisfactory generalization capabilities. We applied random augmentations, including flips, shifts, brightness and contrast adjustments, hue shifts, histogram equalization, and rotation to PH2 data and achieved an IoU of 0.8704, a precision of 0.8801, and a recall of 0.9611 (mean of 10 runs with random augmentations), which is similar to the described test results on the original dataset shown in Table 11. The PH2 dataset was used for the statistical analysis. For the teacher model, we achieved IoU of 0.87 with a 95% confidence interval of [0.84, 0.89], precision of 0.90 with a 95% confidence interval of [0.88, 0.91], recall of 0.94 with a 95% confidence interval of [0.92, 0.95]. For the student* model, we achieved IoU of 0.88 with a 95% confidence interval of [0.85, 0.89], precision 0.89 with a 95% confidence interval of [0.87, 0.90], recall 0.97 with a 95% confidence interval of [0.95, 0.98]. (lines 387-398)

In section 4.3. The robustness of the model:

The problem of the robustness of a model for skin cancer segmentation has been discussed in many publications[50]. It depends on various factors, including its architecture, training data, generalization ability, and performance across different datasets and scenarios.

Augmenting the training data with techniques like transposition, vertical and horizontal flip, brightness, contrast, hue adjustments, CLAHE, shifts, rotation, and coarse dropout improves the model's ability to generalize to unseen data and enhance its robustness.

The results of experiments with test time augmentations on the PH2 dataset, i.e. IoU=0.87, Prec=0.88, Recall=0.96, confirm that the proposed solution maintains high accuracy even with minor variations in the input data.

The metrics proposed in the validation experiment provide insights into the model's performance and help identify potential weaknesses. For the student* model, we notice the increasing precision and recall parameters. Calculated confidence intervals for evaluation metrics to quantify the uncertainty in the model's performance estimates, presented in section 3.4, confirm the high robustness of the proposed solution.

The models were evaluated on unseen data from different sources to assess their generalization ability. The PH2 dataset was used in the final evaluation. Our models generalize well to diverse datasets and imaging conditions beyond the training distribution. The obtained evaluation metrics for the PH2 dataset remain at the same level as the calculated for the ISIC dataset. The low variability of the result suggests good generalization performance. (lines 463 – 484)

Reviewer 2 Report

Comments and Suggestions for Authors

This study proposes a novel approach to semi-supervised skin lesion segmentation based on the Noisy Student training method, aiming to use unlabeled data to improve the results of machine learning models and to improve the quality of skin lesion segmentation, which is important for skin cancer diagnosis. However, the following issues remain to be addressed in this paper:

1. One of the conclusions proposed in this paper is that this simple self-training network can outperform a complex network such as DuAT in the task of skin lesion segmentation. Then why did this study choose to compare the performance of these complex networks? Are there any other networks for comparison? In principle, can you give the relevant reasons for the better performance of this research method compared with these networks?

2. In section 3.4, for the student model with a ResNet34 backbone and simple augmentations, why was better performance on the ISIC 2018 dataset achieved for models with an m = 2 ratio, nonetheless a model with m = 4 was used as a teacher? What is the detailed basis for making such a choice?

3. The flow chart in Figure 1 needs to be further optimized. The relationship between the teacher model, the student model, and the student* model can be further explained in the figure.

4. The first letter of the first paragraph of sections 2.1.1 and 2.1.2 needs to be capitalized.

5. The introduction of each teaching model in the architecture selection for the teacher model in Section 3.1 should be placed in the Materials and Methods in Chapter 2 as an introduction of background methods.

6. The legend of Figure 7, and Figure 8 displays are incomplete.

Author Response

Thank you for taking the time to review our manuscript. We appreciate your accurate comments and suggestions to improve our manuscript. Please find the detailed responses to each comment; see also the attachment for responses in table form.

Comment 1: One of the conclusions proposed in this paper is that this simple self-training network can outperform a complex network such as DuAT in the task of skin lesion segmentation. Then why did this study choose to compare the performance of these complex networks? Are there any other networks for comparison? In principle, can you give the relevant reasons for the better performance of this research method compared with these networks?

Ad. 1 We selected those networks for comparison as they represent state-of-the-art models for skin lesion segmentation. In section 3.1. "Architecture selection for teacher model" (page 10, lines 310-321) we train general-purpose segmentation models to see how they perform for skin lesion segmentation. The general purpose model's performance is worse than SOTA's (Table 1, page 10).  Application of noisy student training results in performance improvement. We plan to investigate the influence of self-training on designed architectures in future work; we suspect that designated architecture will also benefit from the application of self-training.

We changed part of the introduction of the models to "In the study, we used model architectures with a notable position in the literature as we want to focus our research on designing a scalable training approach rather than on deep learning network architecture. We want to separate the influence of noisy student training and application-specific model adjustments." lines (230- 233)

In conclusion, we added, “In the future research, we plan to investigate the influence of self-training on application-specific model architecture.” (lines 498-499)

Comment 2: In section 3.4, for the student model with a ResNet34 backbone and simple augmentations, why was better performance on the ISIC 2018 dataset achieved for models with an m = 2 ratio, nonetheless a model with m = 4 was used as a teacher? What is the detailed basis for making such a choice?

Ad. 2 We used two datasets to test the model ISIC2018 validation subset and ISIC 2017 + ISIC 2018 test subsets. The ISIC 2017+2018 dataset has more samples, and it is more diverse, so we made our decision based on the results on this dataset.

Comment 3: The flow chart in Figure 1 needs to be further optimized. The relationship between the teacher model, the student model, and the student* model can be further explained in the figure.

Ad. 3 We added arrows to Figure 1 (page 6) that show where the pseudo labels are applied.

Comment 4: The first letter of the first paragraph of sections 2.1.1 and 2.1.2 needs to be capitalized.

Ad. 4 We capitalized the first letters of the mentioned sentences (lines 252 and 263).

Comment 5: The introduction of each teaching model in the architecture selection for the teacher model in Section 3.1 should be placed in the Materials and Methods in Chapter 2 as an introduction of background methods.

Ad. 5 We moved the introduction of segmentation models from subsection 3.1 Architecture selection for teacher model to subsection 2.1 Models architectures (lines 229-249).

Comment 6: The legend of Figure 7, and Figure 8 displays are incomplete.

Ad. 6 We added missing axes labels to figures 7, 8, 9, and 10. To increase clarity, we also refined labels for figures 5, 7, 8, 9, 10, 11, and 12. (pages 11-16).

Reviewer 3 Report

Comments and Suggestions for Authors

In this paper, the authors purportedly propose a novel semi-supervised approach for skin lesion segmentation, leveraging self-training with a Noisy Student model to utilize unlabeled data effectively. They claim to report a significant improvement in segmentation performance while reducing reliance on labeled data, achieving high mIoU scores on the ISIC 2018 and PH2 datasets.

While Self-Training and Semi-Supervised Learning are important topics in the medical imaging field, and hence research in this direction is motivated, the authors should do a much better job of studying related works and properly claiming what they are adding to the existing literature body with this work. In lines 180-181, the authors claim no previous use of Noisy Student training in medical image segmentation. Searching on Google Scholar, the papers that cite arXiv:2106.01226 and include <<medical image segmentation>> yield more than 400 results. Similarly, searching <<"noisy student" medical image segmentation>> in Google Scholar leads to more than 1000 papers. Just as an example of recent work, see doi:10.1016/j.bspc.2022.104203. Since this is a major claim for the novelty presented in the paper, the authors should thoroughly investigate the existing literature, properly rewrite the Related Work section, and rectify this statement. Also, the conclusions should be rewritten after proper reconsideration of other works.

Minor comments:

- In Equation (3), the cardinality operator is missing in the denominator.

- Please check the spaces before/after references.

Author Response

Thank you for taking the time to review our manuscript. Please find the detailed responses to each comment; see also the attachment for responses in table form.

We want to highlight that we did not reduce the number of human-labeled data in each training set; we only added new data with labels generated by a model. This technique serves as a regularization of the model.

Comment 1: While Self-Training and Semi-Supervised Learning are important topics in the medical imaging field, and hence research in this direction is motivated, the authors should do a much better job of studying related works and properly claiming what they are adding to the existing literature body with this work. In lines 180-181, the authors claim no previous use of Noisy Student training in medical image segmentation. Searching on Google Scholar, the papers that cite arXiv:2106.01226 and include <<medical image segmentation>> yield more than 400 results. Similarly, searching <<"noisy student" medical image segmentation>> in Google Scholar leads to more than 1000 papers. Just as an example of recent work, see doi:10.1016/j.bspc.2022.104203. Since this is a major claim for the novelty presented in the paper, the authors should thoroughly investigate the existing literature, properly rewrite the Related Work section, and rectify this statement. Also, the conclusions should be rewritten after proper reconsideration of other works.

Ad. 1 Agree; the statement was definitely too broad, as we focused our research strictly on skin lesion segmentation. We rewrote the statements to address only skin lesion segmentation.

A search of << "noisy student" skin lesion segmentation>> returns 112 results, which mainly contain classification methods that either use model backbone trained with a noisy student or use the noisy student technique for improving classification; a portion of the articles only briefly mention noisy student training in the literature review, those articles often apply to different types of medical images like vitiligo or breast lesion. One article ("Network-Agnostic Knowledge Transfer from Learned Representations for Medical Image Segmentation") uses a similar method to transfer knowledge between datasets and uses skin lesions as a transfer dataset. However, it uses a network trained on one type of medical image to predict labels for different types of medical images. 

Therefore, we changed the sentence “We found no previous use of Noisy Student training in medical image segmentation.” to “We found no previous use of Noisy Student training in skin lesion segmentation.” (lines 188 -189)

We also changed the sentence “Our goal is to combine a limited set of labeled data and a large amount of unlabeled data to increase the accuracy and robustness of medical image segmentation.” to “Our goal is to combine a limited set of labeled data and a large amount of unlabeled data to increase the accuracy and robustness of lesion segmentation.” (lines 202-203)

Comment 2: In Equation (3), the cardinality operator is missing in the denominator.

Ad. 2 We added the cardinality operator in Equation 3 (page 8)

Comment 3: Please check the spaces before/after references.

Ad. 3 We fixed inconsistent spaces before and after references.

Round 2

Reviewer 3 Report

Comments and Suggestions for Authors

The authors addressed my comments.